# Negative Thermal Quenching of Photoluminescence: An Evaluation from the Macroscopic Viewpoint

**DOI:** 10.3390/ma17030586

**Published:** 2024-01-25

**Authors:** Shirun Yan

**Affiliations:** Department of Chemistry, Fudan University, Shanghai 200438, China; sryan@fudan.edu.cn

**Keywords:** reliability assessment, negative thermal quenching, consistency of the measurement conditions

## Abstract

Negative thermal quenching (NTQ) denotes that the integral emission spectral intensity of a given phosphor increases continuously with increasing temperature up to a certain elevated temperature. NTQ has been the subject of intensive investigations in recent years, and a large number of phosphors are reported to have exhibited NTQ. In this paper, a collection of results in the archival literature about NTQ of specific phosphors is discussed from a macroscopic viewpoint, focusing on the following three aspects: (1) Could the NTQ of a given phosphor be reproducible? (2) Could the associated data for a given phosphor exhibiting NTQ be in line with the law of the conservation of energy? (3) Could the NTQ of a given phosphor be demonstrated in a prototype WLED device? By analyzing typical cases based on common sense, we hope to increase awareness of the issues with papers reporting the NTQ of specific phosphors based on spectral intensity, along with the importance of maintaining stable and consistent measurement conditions in temperature-dependent spectral intensity measurement, which is a prerequisite for the validity of the measurement results.

## 1. Introduction

The negative thermal quenching (NTQ) of photoluminescence has been the subject of intensive investigations in the past two decades. NTQ refers to the fact that the integral emission spectral intensity of a phosphor gradually increases with an increase in the working temperature within a certain temperature range, so that at a very high temperature (e.g., 200 °C), the intensity of the emission spectrum of the phosphor is the same as or even higher than at low temperature. In some papers, NTQ has been termed as abnormal thermal quenching, zero thermal quenching [1,2,3], anti-thermal quenching [4], etc. NTQ has been reportedly observed in a wide variety of phosphors, such as those singly doped with rare earth ions [1,2,3,4,5,6,7,8,9,10,11,12,13,14,15,16,17,18,19,20,21,22], transition metal ions [23,24,25,26,27,28,29,30,31,32,33,34,35,36], ns^2^ ions (Bi^3+^) [37]; those doubly doped and triply doped with rare earth and transition metal ions [38,39,40,41,42], liquid-crystalline molecules and undoped metal halide [43,44]; as well as in up-conversion phosphor [45,46]. In these papers, the degree of NTQ, that is, the magnitude of the emission spectral intensity enhancement in the phosphor at high temperatures compared to the value at low temperature, ranges from 0.3% [10] to 34,700% [28]. The mechanisms proposed for the NTQ of specific phosphors are also distinct. Many researchers have attributed NTQ to defects in the phosphor, which absorb and store the excitation light at low temperature and transfer the energy to the luminescent center at high temperatures [1,2,3,4,5,6,7,8,9,10,11,12,13,14,15,16,17,18,19,20]. Some researchers have ascribed the NTQ of Mn^4+^-activated fluoride phosphors to an increase in phonon numbers at high temperatures that makes it possible to gain parity and spin-forbidden ^2^*E*_g_ → ^4^*A*_2g_ transitions [26,27,28,29,30,31,32,33,34,35,36], while others believe that the details of the phase transition process in Na_3_Sc_2_(PO_4_)_3_:Eu^2+^ phosphors causes NTQ [6], etc. Some articles have proposed strategies for designing anti-thermal-quenching phosphors based on the mechanism of NTQ [47]. 

One of the motivations for designing thermal quenching (TQ)-resistant phosphors is the application in high-power white-light-emitting diodes (WLEDs) and laser-driven lighting, because the temperature of the phosphors during the operation of WLED and laser-driven lighting devices is relatively high. The reports on NTQ of various phosphors seem to imply that fluorescence TQ, which is a problem hindering the application of phosphors in high-temperature circumstances, has been solved. It is puzzling and regretful that, to the best of our knowledge, there seem to be few commercial WLED manufacturers advertising their WLED products that can demonstrate NTQ of any phosphor at high current densities. At the same time, the validity and reliability of the NTQ mechanisms and experimental results reported in the literature have aroused controversy. Hao et al. pointed out that the so-called NTQ of Sr_8_ZnSc (PO_4_)_7_:Tb^3+^ phosphor, caused by energy transfer from defect levels at different depths in [18], is not convincing, as the luminescence lifetimes of Tb^3+^ at different temperatures in the article do not support this conclusion [48]. Meanwhile, the temperature dependence of the emission spectra of Tb^3+^ in Ba_2_Y_5_B_5_O_17_ (BYBO) and Ba_2_Lu_5_B_5_O_17_ (BLBO) measured by Hao et al. also showed a similar behavior to that of Sr_8_ZnSc(PO_4_)_7_:Tb^3+^. Nevertheless, lattice defects due to aliovalent substitutions were not available in BYBO and BLBO in [48]. The author of this paper also reviewed and commented on the reports about NTQ for Mn^4+^-doped fluoride phosphors and Eu^2+^-doped phosphors. Based on a comprehensive analysis of corresponding experimental data, it was suggested that the mechanisms proposed for NTQ of Na_3_Sc_2_ (PO_4_)_3_:Eu^2+^, caused by the phase transformation details [49] of Mn^4+^-activated fluoride phosphors being due to the increase in phonon numbers at high temperatures [50,51], and of Eu^2+^-activated phosphors stemming from defect-related energy transfer, were unconvincing [52]. It seems problematic for one to attribute NTQ to an intrinsic property of these phosphors. The NTQ phenomenon observed by the researchers is likely a measurement error caused by the volume change arising from the thermal expansion of the phosphor under investigation during the measurement process [49,50,51,52]. The temperature-dependent emission spectral intensity alone seems not to be a reliable measure for evaluating fluorescence TQ, especially for those phosphors having a larger volume change at elevated temperatures due to thermal expansion or phase transition, since emission spectral intensity is affected by not only the quantum efficiency (QE) of a phosphor but also extrinsic factors at elevated temperatures. Only when it is also corroborated by the temperature dependence of the QE and/or lifetime of a given phosphor, or demonstrated in practical applications, could the statement hold that NTQ of a given phosphor is an intrinsic property of the phosphor.

The objective of this paper is to scrutinize typical experimental results in the literature regarding NTQ of specific phosphors from a macroscopic viewpoint, focusing on the following three aspects: (1) Could the NTQ of a given phosphor be reproducible? (2) Could the associated data for a given phosphor exhibiting NTQ be in line with the law of the conservation of energy? (3) Could the NTQ of a given phosphor be demonstrated in a prototype WLED device? By analyzing the typical cases based on common sense, it is hoped to arouse readers’ reflection on and awareness of whether the papers reporting NTQ of specific phosphors are trustworthy. It is also hoped that the stability and consistency of conditions in temperature-dependent spectral intensity measurement could attract more attention in order to avoid misleading or hindering truly meaningful research on anti-TQ phosphors, due to the plethora of papers in the published literature in which NTQ was achieved based largely on the crude and unreliable temperature-dependent emission spectral intensity of specific phosphors.

## 2. Assessment of the Selected Literature Results

### 2.1. Could NTQ of a Given Phosphor Be Well Reproducible?

It is generally acknowledged that an essential requirement for experimental results on the material properties of scientific significance that deserve discussion is repeatability. If NTQ is an intrinsic property of a specific phosphor, the temperature dependence of the emission intensity of a given phosphor measured by different researchers or by the same research group at different times should be reproducible within a reasonable range of uncertainty.

Na_3_Sc_2_(PO_4_)_3_:Eu^2+^, a sodium superionic conductor (NASICON)-type phosphor, is one of the most heavily investigated Eu^2+^-doped phosphors. The TQ property of Na_3_Sc_2_(PO_4_)_3_:Eu^2+^ was reported by more than five independent research groups during 2016–2022 [1,2,5,6,7,8,42]. In particular, the paper authored by Im et al. entitled “A zero-thermal-quenching phosphor”, published in “Natural Materials” in 2017, has been widely cited in reports on NTQ [2]. These authors all claimed that the phase-pure Na_3−2x_Sc_2_(PO_4_)_3_:xEu^2+^ phosphors had an NTQ phenomenon. Nevertheless, both the temperature at which the maximum emission intensity is obtained and the magnitude of the maximum emission intensity with respect to the intensity at room temperature, i.e., *I*_max_(*T*), for the Na_3_Sc_2_(PO_4_)_3_: Eu^2+^ phosphor with the same Eu^2+^ doping concentration reported by different research groups or even by the same research group in different publications, are considerably different. Wang et al. [5] reported that, while heating the phosphor from 25 to 250 °C, the emission intensity increases gradually at first and then reaches the maximum at 150 °C, upon which the emission intensity starts to decrease. The emission intensity of the phosphor with an optimum concentration of Eu^2+^ doping Na_3_Sc_2_(PO_4_)_3_:0.03Eu^2+^ (NSP:0.03Eu^2+^) at 150 °C is 110% compared to that at room temperature, as shown in Figure 1a [5]. Im and co-workers reported that the Na_3_Sc_2_(PO_4_)_3_:0.03Eu^2+^ (NSPO:0.03Eu^2+^) phosphor shows a ~25% increase in emission intensity in the β-phase (above 65 °C) compared to the room-temperature emission, and it reaches a maximum at ~164 °C (β-phase) [2]. Im and co-workers reported recently in another paper that the emission intensity of the Na_3_Sc_2_(PO_4_)_3_:0.03Eu^2+^ increases by about 70% when the temperature is increased from 25 to 150 °C and reaches a maximum at about 175 °C, as displayed in Figure 1b [1]. Xian et al. [8] reported that the emission intensity of Na_3_Sc_2_(PO_4_)_3_:0.03Eu^2+^ under 340 nm excitation increases when increasing the temperature from 300 up to 425 K. Further increasing the temperature leads to the TQ phenomenon. The emission intensity at 425 K is ~120% compared to the emission intensity at room temperature, as depicted in Figure 1c [8]. Liu et al. [42] reported that the integrated luminescence intensity of Na_3_Sc_2_(PO_4_)_3_:0.03Eu^2+^ (NSPO:0.03Eu^2+^) increases by ~30% when the temperature is increased from 30 to 150 °C, as illustrated in Figure 1d [42]. Yan et al. [7] reported that the integrated emission intensity of Na_3_Sc_2_(PO_4_)_3_:0.03Eu^2+^ (NSP:0.03Eu^2+^) increases with increasing temperature from 300 up to 425 K and then decreases when further increasing the temperature above 425 K. The emission intensity at 425 K is ca. 160% compared with the initial emission intensity at room temperature, as shown in Figure 1e [7].

Mn^4+^-doped fluorides are a kind of red-emitting phosphor, which have been extensively reported to have NTQ [25,26,27,28,29,30,31,32,33,34,35,36], although many researchers have reported a normal TQ for Mn^4+^ luminescence [53,54,55,56,57,58]. Taking the most extensively studied phosphor, K_2_SiF_6_:Mn^4+^, as an example, some researchers reported that the K_2_SiF_6_:Mn^4+^ shows a normal TQ behavior at high temperatures, analogous to that of most inorganic phosphors [53,56,57,58]. On the contrary, many researchers reported that the K_2_SiF_6_:Mn^4+^ phosphor shows an NTQ behavior [25,26,27,28,29,30,31,32]. However, both the temperature at which the maximum emission intensity observed and the magnitude of the maximum emission intensity with respect to the intensity at room temperature of phase-pure K_2_SiF_6_:Mn^4+^ phosphor reported by different researchers, or even the same commercial TriGain^®^ K_2_SiF_6_:Mn^4+^ phosphor measured by the same research group [25,59], are distinct, as illustrated in Figure 2. The author discussed this issue in a previous paper [51].

Based on the results in Figure 1 and Figure 2, it appears hard for one to draw a conclusion that the NTQ phenomenon of either Na_3_Sc_2_(PO_4_)_3_:0.03Eu^2+^ or K_2_SiF_6_:Mn^4+^ phosphor is reproducible within a reasonable range of measurement errors.

### 2.2. Could the Associated Data of a Specific NTQ Phosphor Be in Line with the Law of Conservation of Energy?

As is well known, phosphors are a kind of frequency (or wavelength) conversion material. Down-shifting phosphors can convert one short-wavelength (high-frequency) photon to one long-wavelength (low-frequency) photon. Due to the possible nonradiative decay process of the excited state, part of the absorbed energy could be lost in the phosphor as heat. The law of the conservation of energy states that when light is absorbed by a phosphor, the energy must go somewhere. Considering that no afterglow luminescence has been observed at any temperature in the phosphors reported with NTQ, it means that the light absorption and emission processes do not involve energy storage and a delayed release for these phosphors. Therefore, the sum of the instantaneously emitted energy (*E_Em_*) by a phosphor and the energy lost in the phosphor should be equal to the absorbed energy (*E_Abs_*) by the phosphor. The energy lost in the phosphor includes the energy lost due to nonradiative decays (*E_NR_*) and Stokes shift (*E_SS_*). That is, the flow of optical energy passing through a phosphor in a unit time period at any temperature could be expressed by the following equation:*E_Abs_* = *E_Em_* + *E_NR_* + *E_SS_*(1)

Let us first discuss the left-hand side of energy balance, Equation (1), for a phosphor (the absorbed energy by a phosphor (*E_Abs_*)). Considering that a monochromatic radiation is usually chosen as an excitation light (*λ_exc_*) when measuring an emission spectrum, the absorbed energy by the phosphor can be approximated as the product of the energy of single-frequency excitation radiation and the total number of photons absorbed. The number of photons absorbed by the phosphor is proportional to the number of activator centers per unit volume (*N*) and the transition probability (*P_if_*). According to the literature, the electric dipole absorption probability of a two-level center in crystal could be expressed as [60]:(2)$$P_{if}=\frac{\pi }{3n\varepsilon_{0}c_{0}\hbar^{2}}I{\left|\mu_{if}\right|}^{2}{\left(\frac{E_{loc}}{E_{0}}\right)}^{2}\delta \left(\Delta \omega \right)$$
where $I=\frac{1}{2}nc_{0}\varepsilon_{0}E_{0}^{2}$ is the intensity of the incident radiation (assuming an incident plane wave), *ℏ* is the reduced Planck’s constant, *c_0_* is the speed of light in a vacuum, *n* is the refractive index of the phosphor, *ε*_0_ is the permittivity in a vacuum, *μ_if_* is the matrix element of the electric dipole moment, and *E_loc_* and *E_0_* are the actual local electric field acting on the valence electrons of the absorbing center due to the electromagnetic incoming wave and the average electric field in the medium, respectively. *δ*(∆*ω*) is the frequency of the incident monochromatic radiation [60]. Based on Equation (2), one can hardly expect that the absorption probability increases with increasing temperature.

In fact, the temperature dependences of the absorption intensity of specific phosphors that exhibited NTQ have been investigated by some researchers. Shao et al. investigated temperature-variable diffuse reflection spectra of the K_2_SiF_6_:Mn^4+^ phosphor in a temperature range from 20 to 80 °C [27] and found no obvious changes in absorption rates when increasing the temperature. Im et al. studied the temperature-dependent absorption fraction of Na_3−2x_Sc_2_(PO_4_)_3_:xEu^2+^ (x = 0.01, 0.03, 0.07) under 370 nm excitation in the temperature range of 25—175 °C, with a temperature interval of 25 °C. The results showed that the absorption fraction of the Na_3−2x_Sc_2_(PO_4_)_3_: xEu^2+^ phosphors remained unchanged with rising temperature, suggesting that the enhanced emission intensity does not arise from the increase in the absorption fraction [2].

Next, we discuss the first item on the right-hand side of Equation (1), the emitted energy by the phosphor (*E_Em_*), which is called radiant power or radiant flux, measured in joules per second or watts in radiometry. It is easily imagined that *E_Em_* is associated with the emission spectral intensity. Given that in spectral measurement, the emission by the phosphor upon excitation is directional–hemispherical while the spectrum recorded by a spectrofluorometer is the radiant power per unit solid angle of a specific direction determined by the geometrical configurations of the spectrofluorometer (radiant intensity), the total radiant power can be obtained by 2π steradian times the radiant intensity [61]. If the emission spectrum is plotted as emitted energy per constant wavelength interval, then
(3)$$E_{\mathrm{Em}}=2\pi \int_{380}^{780}I(\lambda )d\lambda$$
where *I*(*λ*) is the spectral intensity, which is defined as the radiant intensity per unit wavelength interval.

The emitted energy by a phosphor is determined by the number of activator centers per unit volume in the excited state (*N**), the probability of radiative transition of the excited electron (*P_fi_*), and energy difference between the emitting levels of excited and ground states [62,63].

The second term on the right-hand side of Equation (1), the energy lost in the phosphor due to nonradiative decay (*E_NR_*), can be quantified by internal quantum efficiency (*IQE*), defined as the ratio of the number of emitted quanta to the number of absorbed quanta by the phosphor. The *IQE* is determined by the probability of radiative and nonradiative decays of the excited center per unit time (i.e., decay rates) and can be expressed as follows:(4)$$IQE=\frac{\Gamma_{R}}{\Gamma_{R}+\Gamma_{NR}}$$
where *Γ_R_* and *Γ_NR_* are the radiative and nonradiative decay rates of the excited center, respectively [26,29]. Equation (4) indicates that *E_NR_* increases when the *IQE* of a phosphor decreases, and vice versa.

The third term on the right-hand side of Equation (1), *E_SS_* stems from the interaction between electrons in the excited state and crystal lattice vibrations. Upon excitation from the ground state, the excited electron quickly relaxes into the lowest vibrational level within the excited electronic state, losing some of the initial excitation energy as heat. Subsequently, the electron at the lowest vibrational level of the excited state decays to a ground state accompanied with light emission. The vibrational relaxation of the excited electron results in the energy of emitted photons will be less than that of absorbed photons [62,63,64].

The emission spectral intensity of a phosphor increases with increasing temperature, i.e., NTQ, suggesting that the emitted energy (*E_Em_*) by the phosphor increases with increasing temperature. If the absorbed energy does not increase, in order to ensure that Equation (1) holds, the second or/and third terms on the right-hand side of the formula should decrease in an equal proportion. The third term, Stokes loss, is caused by the interaction between the excited-state electron and lattice vibrations. As the temperature increases, lattice vibrations strengthen. It seems unlikely that the Stokes loss could decrease with increasing temperature. For Eu^2+^- and Ce^3+^-activated phosphors, the Stokes shift increases with increasing temperature due to an enlarged activator site induced by lattice thermal expansion along with enhanced vibrations at high temperatures [65]. Therefore, it appears that only when the energy lost in the phosphor due to nonradiative decay (*E_NR_*) decreases, that is, the IQE increases with increasing temperature, could Equation (1) hold.

In order to increase IQE of a given phosphor with increasing temperature, it is necessary to increase the radiative decay rate (*Γ_R_*) with increasing temperature, or decrease the nonradiative decay rate (*Γ_NR_*) with increasing temperature, or both occur simultaneously. Nevertheless, for the allowed electric dipole transitions, like 4f^n−1^5d^1^ →  4f^n^5d^0^ transitions of Eu^2+^ and Ce^3+^-doped phosphors, the radiative decay rate (*Γ_R_*) is constant, independent of temperature, and it is hard to imagine that the radiative decay rate (*Γ_R_*) for these phosphors could increase with increasing temperature. At the same time, the nonradiative decay rate (*Γ_NR_*) commonly increases above a critical temperature, leading to luminescence TQ. The temperature dependence of the nonradiative decay rate can be expressed as [62,63]:*Γ_NR_* = *A* exp(−Δ*E*/*k_B_T*) (5)
in which *A* is a constant (units s^−1^), Δ*E* is the energy barrier for thermal quenching, and *k_B_* is the Boltzmann’s constant (8.617 × 10^−5^ eV.K^−1^). Irrespective of the mechanism (or pathway) for nonradiative decay being either multi-phonon emission, crossover of the ground and excited potential curves in the configuration coordinate diagram, thermal-assisted energy transfer to defects, or photoionization of the 5d electron into the conduction band of the matrix, it is unlikely that the nonradiative decay rate decreasing with increasing temperature could happen based on Equation (5).

There are also reports in the literature on the temperature dependence of IQE of some phosphors that showed NTQ in the emission spectral measurements. Zhou et al. reported that the peak intensity and integrated area intensity of SB_0.3_PE at 150 °C reached 108% and 124% of the room-temperature value, respectively, as illustrated in Figure 3a [11]. The IQE of the phosphor at 150 °C decreased to 85% from the room-temperature value 100%, and the absorption of the excitation light within the range of 25–150 °C was almost constant, as shown in Figure 3b [11]. Figure 3 shows that although the emission spectral intensity increases with increasing temperature, neither the energy absorbed by the phosphor nor the IQE of the phosphor increase with increasing temperature, indicating that a decrease in the energy loss due to nonradiative decay has not been observed in this phosphor that exhibited NTQ.

There are also reports in the literature on the temperature-dependent QE of K_2_SiF_6_:Mn^4+^, as exemplified in Figure 4 [53]. It can be seen from Figure 4 that both the IQE and EQE of the phosphor remain almost unchanged with the increase in temperature from room temperature to 150 °C, indicating that the energy lost in the phosphor due to nonradiative decay does not decrease with the increase in temperature, and the absorption of the excitation light by the phosphor does not increase with the increase in temperature either.

The results in Figure 3 and Figure 4 reveal that although the *E_Em_* increases with increasing temperature (NTQ), *E_Abs_* and *E_NR_* remain almost constant with temperature, considering that *E_SS_* could hardly decrease with increasing temperature, suggesting that the relation between these data could not be in line with Equation (1).

### 2.3. Could NTQ of a Given Phosphor Be Demonstrated in Prototype WLED Device?

The aim of phosphor research lies not only in exploring the underlying theoretical issues but also in finding practical applications for specific phosphors. If a phosphor possesses a peculiar attribute, in addition to be able to prove through characterizations, it should also be demonstratable in practical applications. One of the motivations for designing TQ-resistant phosphors is applications in high-power WLED and/or laser-driven lighting, because the temperature of the phosphors during the operation of the WLED or laser-driven lighting devices is relatively high [64]. If the QE of the phosphors decreases at the working temperature, the efficiency and color of the device will change correspondingly. Generally, four metrics—luminous efficacy, color-rendering index (CRI), correlated color temperature (CCT), and lifetime—are used to describe the performance of WLED devices [64]. The luminous efficacy expressed in lumens per watt is a parameter describing how bright the radiation is perceived by the average human eye. It scales with the eye sensitivity curve *V*(λ) and can be calculated from the emission spectrum *I*(λ). As the average human eye sensitivity peaks at a wavelength of 555 nm, the fraction of green-yellow light, along with its wavelength and QE, has the greatest impact on luminous efficacy of WLEDs. The CRI quantifies the color-rendering ability or color reproducibility of a white-light source scored on a scale from 0 (no color reproducibility) to 100 (perfect reproducibility, achieved by black body radiators). The CCT describes the hue of the white light. The lower CCT value represents warmer light (being yellower), while the higher CCT value denotes cooler light (being bluer). Generally, the CCT of a WLED device is largely determined by the fractions of red and blue light in the white spectrum. An increase in CCT means that either the fraction of the blue light in the white spectrum increases or the fraction of the red light decreases. On the contrary, a decrease in CCT means that the fraction of the red light in the white spectrum increases, or the fraction of the blue light decreases [64,66]. Given that the main concern of this paper is whether the phosphors that have exhibited NTQ in temperature-dependent spectral measurement could demonstrate a similar performance in a prototype WLED device, along with that the emitting colors of the phosphors discussed in this paper, which reportedly exhibited NTQ of blue and red, respectively, we focus our discussion on the variation in CCT of the prototype WLED device at different drive currents.

Table 1 compiles the results in the published literature about chromaticity coordinates and the CCT of four prototype WLED devices, which were fabricated by coating an InGaN blue chip or a UV chip using the phosphor blend containing specific phosphors, which showed NTQ in the spectral measurement under different drive currents. No. 1 and No. 2 devices are composed of a blue chip and the phosphor blend, and the red-emitting phosphors K_2_TiF_6_:Mn^4+^ [26,29,30] and K_3_AlF_6_:Mn^4+^ [67,68,69] have been reported by multiple independent research groups to exhibit NTQ in the spectral measurements. No. 3 and No. 4 devices are composed of a UV chip and the phosphor blend, and the blue-emitting phosphors Na_3_Sc_2_(PO_4_)_3_:0.03Eu^2+^ (NSPO:Eu^2+^) and Sr_1.38_Ba_0.6_P_2_O_7_:0.02Eu^2+^ (SB_0.3_PE) have been reported by multiple independent research groups to exhibit NTQ in the spectral measurements [1,2,5,6,7,8,11]. Could the red-emitting phosphors and/or blue-emitting phosphors also exhibit NTQ in the respective WLED device?

As discussed earlier, if the emission intensity of red-emitting phosphor increases with temperature while the emission intensity of blue-emitting phosphor decreases with increasing temperature (normal TQ), the CCT of the WLED is expected to decrease at elevated temperatures. If the emission intensity of blue-emitting phosphor increases with temperature while the emission intensity of red-emitting phosphor decreases with temperature, the CCT of the WLED is expected to increase at high temperatures. If the quenching rate with the temperature of the blue-emitting and red-emitting phosphors is the same, the CCT of the WLED changes very little with temperature.

Table 1 shows that the CCT of No. 1 WLED increases gradually when the drive current increases from 20 to 120 mA, while the CCT of No. 2 WLED increases steadily with an increasing drive current from 40 to 240 mA. Further increasing the drive current from 240 to 300 mA, the CCT of No. 2 WLED decreases. The variation in CCT with drive current of these two WLED devices indicates that the fraction of the red light in the whole spectrum of these two WLED decreases with increasing drive current below 240 mA, implying that NTQ of the respective red-emitting phosphors, i.e., the emission intensity increasing with temperature, could not be substantiated by its performance in these two WLED devices.

It is worth mentioning that, in addition to TQ, excitation saturation could also result in a decrease in luminescence efficiency (droop) of a specific phosphor at a high drive current of WLED. The excitation saturation refers to the fact that the luminescence intensity of a phosphor does not linearly increase with an increase in excitation power. The excitation saturation is generally caused by ground-state depletion and associated with the luminescence lifetime of the phosphor [64,71,72]. The luminescence lifetime of Mn^4+^ -activated phosphors (on the order of milliseconds at room temperature) is longer than that of Ce^3+^-activated phosphors (in the order of nanoseconds) and Eu^2+^-activated phosphor (in the order of microseconds) [26,52,73]. The experimental results showed that Mn^4+^-activated fluorides had a considerably lower threshold for excitation saturation than that of Eu^2+^ and Ce^3+^ -activated phosphors in WLEDs [58,74,75]. It is not clear which drive current is the onset of excitation saturation for these two red-emitting phosphors in No. 1 and No. 2 WLED devices and to what extent the excitation saturation also contribute the increase in the CCT of No. 1 and No. 2 WLED devices at a high drive current, given that the blue light in these two WLEDs is provided by an InGaN chip, which also suffers from the external quantum efficiency drop when increasing the drive current above a critical value [76].

The data in Table 1 show that the CCT of No. 3 WLED gradually increases from 7041 K to the maximum 7121 K when the drive current increases from 100 mA to 300 mA. The CCT monotonically decreases with further increasing the drive current above 400 mA, indicating that the fraction of the blue light in the WLED spectrum decreases with increasing drive current above 400 mA; that is, the NTQ of the blue-emitting phosphor Na_3_Sc_2_(PO_4_)_3_:0.03Eu^2+^ could scarcely be substantiated by its performance in the WLED device above 400 mA. If the blue-emitting phosphor has NTQ, i.e., emission intensity increasing with increasing temperature, the CCT of the WLED should increase as well [64]. Given that the blue-emitting phosphor Na_3_Sc_2_(PO_4_)_3_:0.03Eu^2+^ and red-emitting phosphor (SrCa)AlSiN_3_:Eu^2+^ in No. 3 WLED contain the same activator (Eu^2+^), which has a similar luminescence lifetime, excitation saturation could not be the issue resulting in the decrease in CCT with increasing drive current in No. 3 WLED.

In addition to luminescence TQ and excitation saturation, another possible reason that could cause a decrease in the fraction of the blue light under a high drive current is an increased reabsorption process of the other additive phosphor components. In the Supplementary Information in Ref. [2], Im et al. mentioned that the excitation spectrum of the yellow-emitting La_3_Si_6_N_11_:Ce^3+^ and red-emitting (SrCa) AlSiN_3_:Eu^2+^ overlaps with the emission spectrum of Na_3_Sc_2_(PO_4_)_3_:0.03Eu^2+^ phosphor, and there is a significant absorption of the blue emission intensity of Na_3_Sc_2_(PO_4_)_3_:0.03Eu^2+^ phosphor from the EL spectra of the WLED device by the yellow-emitting La_3_Si_6_N_11_:Ce^3+^ phosphor and/or red-emitting (SrCa)AlSiN_3_:Eu^2+^ phosphor, resulting in a decrease in the blue component during WLED fabrication [2]. If the absorption of blue light by the red-emitting or/and yellow-emitting phosphor is enhanced at high temperatures, even if the blue-emitting phosphor Na_3_Sc_2_(PO_4_)_3_:0.03Eu^2+^ has no TQ at a high drive current, an enhanced reabsorption by the red- and/or yellow-emitting phosphor could also result in a decrease in the fraction of the blue component in the WLED device. Then, could the decrease in the CCT of No. 3 WLED device under a high operating current be due to an enhanced reabsorption of the blue light by the red- or/and yellow-emitting phosphors?

There are few reports in the literature on the temperature-dependent excitation spectra of (SrCa) AlSiN_3_:Eu^2+^ and La_3_Si_6_N_11_:Ce^3+^ phosphors to our knowledge. As mentioned earlier, the absorption intensity of Eu^2+^ and Ce^3+^-doped phosphors is determined by the number of activator centers per unit volume and the transition probability from the ground state to the excited state. On the one hand, multiple independent researchers have confirmed that both the yellow-emitting phosphor La_3_Si_6_N_11_:Ce^3+^ [77] and the red-emitting phosphors CaAlSiN_3_:Eu^2+^ and (SrCa)AlSiN_3_:Eu^2+^ [78,79] exhibit TQ, and the mechanism of TQ is generally believed to be the thermally assisted photoionization of 5d electrons of the activator ions (Ce^3+^, Eu^2+^), suggesting that the number of the activator centers per unit volume could hardly increase with increasing temperature. On the other hand, the light absorption of (SrCa) AlSiN_3_: Eu^2+^ and La_3_Si_6_N_11_:Ce^3+^ phosphors originates from the parity-allowed 4f^n^5d^0^ → 4f^n−1^5d^1^ electric dipole transitions of respective Eu^2+^ and Ce^3+^ ions, whose transition probability is expressed by Equation (2) and are independent of temperature. In terms of either the number of activator centers in unit volume or transition probability, it seems difficult to predict theoretically that the absorption of the blue light by the yellow-emitting and/or red-emitting phosphors could increase with increasing temperature.

Multiple independent research groups reported on the absorption spectra of Y_3_Al_5_O_12_:Ce^3+^ (YAG:Ce^3+^) [80,81,82] and Y_3_Ga_5_O_12_:Ce^3+^ (YGG:Ce^3+^) [83] phosphors at different temperatures, as shown in Figure 5a,b. YAG:Ce^3+^ and YGG:Ce^3+^ show two intense bands within the measured range of absorption spectra, which can be attributed to the transition from ground state (4f^1^) to the two lowest-lying 5d^1^ states of Ce^3+^ ion. As the temperature increases, the intensity of the absorption band in the blue region corresponding to the transition from the 4f^1^ to the lowest-lying 5d^1^ state of Ce^3+^ ion decreases gradually, suggesting unambiguously that the absorption of blue light decreases with increasing temperature in the temperature range investigated [80,81,82,83]. The excitation spectrum of La_3_Si_6_N_11_:Ce^3+^ shown in Figure 5c [84] resembles the excitation spectra of YAG:Ce^3+^ and YGG: Ce^3+^. There are also two absorption bands in the blue and nUV regions, corresponding to the transition from ground state of Ce^3+^ ion (4f^1^) to the two lowest-lying 5d^1^ states. It seems reasonable to infer that the temperature dependence of the excitation (absorption) spectrum of La_3_Si_6_N_11_:Ce^3+^ phosphor should be similar to that of YAG:Ce^3+^ and YGG:Ce^3+^; that is, the absorption intensity of the blue light should not increase with increasing temperature.

The temperature-dependent excitation spectra of red-emitting phosphor CaAlSiN_3_:Eu^2+^ were reported by Chen et al., as shown in Figure 6 [85]. It is evident that the excitation intensity of CaAlSiN_3_:Eu^2+^ phosphor by the blue light does not increase with increasing temperature from 20 to 300 K. It seems reasonable to suggest that the variation in the excitation spectrum of (SrCa)AlSiN_3_: Eu^2+^ phosphor with temperature should have a similar trend to that of CaAlSiN_3_:Eu^2+^. The results in Figure 5 and Figure 6 indicate that the decrease in the CCT of No. 3 WLED device at a high drive current seems unlikely to be due to an increased absorption of the blue light by the yellow-emitting phosphor La_3_Si_6_N_11_:Ce^3+^ or/and red-emitting phosphor (SrCa) AlSiN_3_:Eu^2+^ at high temperatures.

Given the above discussion, it appears that attributing the decrease in CCT with increasing drive current for No. 3 WLED device in Table 1 to either an excitation saturation or an increased reabsorption process of the other additive phosphor components is unconvincing. The reason for the decrease in CCT of the WLED device at a high drive current should be that the blue-emitting phosphor Na_3_Sc_2_(PO_4_)_3_:0.03Eu^2+^ suffers from severer fluorescence TQ than that of the red-emitting phosphor. This indicates that the NTQ of the blue-emitting phosphor Na_3_Sc_2_(PO_4_)_3_:0.03Eu^2+^ could not be proven by its performance in the prototype WLED device under a high drive current.

Sr_1.38_Ba_0.6_P_2_O_7_:0.02Eu^2+^ (abbreviated as SB_0.3_PE) is also a blue-emitting phosphor, with an emission maximum at a wavelength of 420 nm under nUV excitation, which reportedly exhibited NTQ. The temperature-dependent emission spectral measurements showed that the peak intensity and integrated area intensity at 150 °C reached 108% and 124% of the room-temperature values, respectively, as shown in Figure 3a [11]. Could the NTQ of Sr_1.38_Ba_0.6_P_2_O_7_:0.02Eu^2+^ be demonstrated in the prototype WLED device?

No. 4 WLED device listed in Table 1 was fabricated by coating a 365 nm nUV chip with the phosphor blend containing the blue-emitting (Sr_1.38_Ba_0.6_P_2_O_7_:0.02Eu^2+^) (SB_0.3_PE), yellow-emitting (SrBa)_2_SiO_4_:Eu^2+^, and red-emitting (SrCa) AlSiN_3_:Eu^2+^ phosphors. The drive-current-dependent CCT of WLED shows that the CCT of No. 4 WLED monotonically decreases from 3831 K to 3775 K when the drive current increases from 25 mA to 200 mA, suggesting that the fraction of the blue-light component in the white spectrum gradually decreases with the increase in the drive current. The decrease in the fraction of the blue-light component in the white spectrum can also be seen from the variation in intensity with drive current for the three phosphors in the blend shown in Figure 7. Based on an analogous reason for the decrease in CCT with increasing drive current in No. 3 WLED, it seems reasonable to deduce that the decrease in the blue-light fraction in No. 4 WLED as the drive current increases could hardly be ascribed to either the excitation saturation or absorption by the yellow-emitting phosphor (SrBa)_2_SiO_4_:Eu^2+^ and/or red-emitting phosphor (SrCa) AlSiN_3_:Eu^2+^. The most likely reason for the decrease in the fraction of the blue-light component in the white spectrum is that the blue-emitting phosphor Sr_1.38_Ba_0.6_P_2_O_7_:0.02Eu^2+^ suffers from severer TQ than that of the red-emitting phosphor (SrCa)AlSiN_3_:Eu^2+^. The fact that the blue-emitting phosphor Sr_1.38_Ba_0.6_P_2_O_7_:0.02Eu^2+^ suffers from severer TQ than that of the red-emitting phosphor (SrCa)AlSiN_3_:Eu^2+^ suggests that the NTQ of the blue-emitting phosphor Sr_1.38_Ba_0.6_P_2_O_7_:0.02Eu^2+^(SB_0.3_PE) could not be demonstrated in a high-power WLED device.

## 3. Discussion

Based on the above discussion, it appears reasonable to note that in terms of the reproducibility of the experimental results, associated data being able to comply with the law of conservation of energy, or demonstrability in practical applications of prototype WLED devices, the NTQ of these phosphors, despite having been extensively reported by numerous researchers and heavily cited in the literature, could hardly be substantiated. The key reason for this predicament, in my opinion, is that the NTQ reported in the literature is based largely on the temperature-dependent emission spectral intensity of a specific phosphor. However, the temperature-dependent emission spectral intensity is not a reliable measure for evaluating luminescence TQ [52]. The contribution of the change in geometrical configuration to the emission spectral intensity at elevated temperatures has been ignored.

It is generally acknowledged that QE is a critical parameter for evaluating the performance of phosphor materials, as it indicates their ability to absorb and convert photons to longer wavelengths. IQE and EQE are two commonly used metrics. To evaluate the quality of a given phosphor at room temperature, one should measure the IQE rather than the absolute emission spectral intensity of the phosphor because a conventional spectrofluorometer can detect only a certain fraction of the emitted light. The size of this fraction depends on many factors, including the numerical apertures for excitation and the solid angle for detection, the emission wavelength, the emission anisotropy, the scattering of the sample and the sample geometry, sensitivity and linearity of the detector, etc.; thus, it is impossible to quantify. A recent work by Song et al. demonstrated that even using a commercial spectrofluorometer under the same conditions (slit width, dwell time, scan range, and sample position) and operated by the same experimenter, the different quartz lids pairs of the sample chamber resulted in an integrated emission spectral intensity variance of more than 10% for the yellow- and red-emitting phosphors recorded by a PMT (photomultiplier tube) of the spectrofluorometer [86]. Therefore, investigation of an IQE of a phosphor by comparing integral emission spectral intensities of the sample and the reference measured using a spectrofluorometer has scientific significance only when the emission spectra are obtained under rigorously identical measurement conditions for the samples of known absorption factors at the excitation wavelength [87]. Maintaining the consistency and stability of the measurement conditions is crucial for the accuracy and reliability of the emission spectral intensity.

However, it is a great challenge to maintain the consistency and stability of the measurement conditions, especially the surface morphology of the sample and geometrical configuration of the spectrofluorometer at different temperatures when measuring the temperature-dependent emission spectral intensity using a conventional spectrofluorometer equipped with a temperature-control accessory. The main reasons are as follows [51,52]:(1)The measurement of temperature-variable emission spectra from room temperature to a specific higher temperature (e.g., 200 °C) involves multiple sample temperature-raising and temperature-holding steps, and, hence, is a time-consuming process. Maintaining the stability of the excitation source power, detector sensitivity, and other measurement conditions without any fluctuation throughout the whole process is a challenge.(2)In the majority of cases, the phosphor expands with increasing temperature, since an increase in energy results in an increase in the equilibrium spacing of the atomic bonds. The optical quartz lid of the sample holder (or sample chamber) for masking phosphor samples is an elastic material that could hardly prohibit the phosphor sample from expanding upon heating at ambient pressure. Given the fact that the homogeneity in chemical composition and/or uniformity in particle size of phosphor samples prepared by laboratory-scale solid-state reactions could hardly be ensured, along with that the phosphor powders are randomly packed in the sample holder, the thermal expansion (volume change) of the phosphor undoubtedly leads to changes in the state of the sample in directions parallel to and perpendicular to the sample surface. On the one hand, the volume change in the sample in the direction parallel to the surface may cause a change in the state of the sample (packing density and flatness) within the cross-sectional area of the incident beam, leading to a change in the reflection/absorption ratio of the phosphor sample. Consequently, a change in the emission spectral intensity is expected. On the other hand, the volume change in the direction perpendicular to the sample surface could affect the distances from the excitation source to the sample surface and from the sample surface to the detector, even though the excitation source, sample holder, and detector of the spectrofluorometer remain static at fixed positions at elevated temperatures. As a consequence, the incident excitation intensity on the phosphor and the emission intensity received by the detector will be biased due to the inverse square law. The magnitude of these changes is associated with the thermal expansion coefficient (lattice rigidity) of a given phosphor, the volume of the sample chamber, the particle size, and packing density of the phosphor could hardly be quantified [51]. Either a change in sample state and surface flatness within the cross-sectional area of the incident excitation light or a change in geometrical configurations of the spectrofluorometer means that the measuring conditions at elevated temperatures have been changed considerably from those at low temperature. Hence, the emission spectral intensity obtained is unable to accurately characterize the variation in phosphor performance any more. Regardless of the volume change in the phosphor being caused by thermal expansion, negative thermal expansion, or phase transformation [1,2,6], the change in crystal cell volume of a phosphor at elevated temperatures makes it difficult to maintain the consistency and stability of the measuring conditions in temperature-variable emission spectral measurements.(3)Given that the homogeneity in chemical composition and/or uniformity in particle size of phosphor samples prepared by laboratory-scale solid-state reactions could not always be ensured [88], the surface of the phosphor sample for spectral measurement is generally not an ideal Lambertian surface. That is, an isotropic radiance in all directions of the hemisphere above the sample surface could not always be ensured. A change in the surface state originating from the volume change due to thermal expansion or negative thermal expansion of the phosphor may result in a change in the scattering of the emission in different directions at elevated temperatures. As a result, the spectral intensity, which records the emitted energy in the specified direction of the spectrofluorometer, may change with the temperature dependence of anisotropic scattering of the phosphor sample, irrespective of whether or not the emitted intensity changes with temperature.(4)Due to thermal expansion or negative thermal expansion of the phosphor sample, the length of the atomic bonds between the activator ion and the ligands will change accordingly. For some phosphors, the energetic positions of the transitions from the ground state to the excited states are extremely sensitive to the coordination field of the activator ion (e.g., 4f^n^5d^0^ → 4f^n−1^5d^1^ transition of Ce^3+^ and Eu^2+^ ions, ^4^A_2_ → ^4^T_2_ transition of Mn^4+^, charge transfer from the ligand to Eu^3+^, etc.). The change in the length of the atomic bonds between the activator ion and the ligands may lead to a shift in the excitation band at high temperatures from that at low temperature, resulting in a change in the absorption of monochromatic excitation light [51,52].

As discussed earlier, the key reason for emission spectral intensity at room temperature unable to be an interlaboratory comparable quantity is that the emission spectral intensity measured is only a certain fraction of the emitted light, and the size of this fraction depends on many factors. It is impossible to keep these factors and measuring conditions quantitively identical and consistent universally. The above discussion suggests that during the measurement of variable temperature emission spectra of phosphors using a commercial spectrofluorometer by an experimenter, the consistency and stability of measuring conditions, especially the surface state (density and flatness) within the cross-sectional area of the incident beam and measuring geometry (source–sample distance, sample–detector distance, and angular distribution of emission), could hardly be ensured, especially for those phosphors that have considerable volume changes due to a thermal expansion, negative thermal expansion, or phase transformation process. In this case, the measurement errors originating from the volume change make the emission spectral intensity measured at elevated temperatures considerably biased from the emitted intensity by the phosphor. Hence, the comparison of emission spectral intensity at higher temperatures with respect to the value at low temperature could not truly characterize the change in the light conversion efficiency of the given phosphor.

Multiple independent research groups investigated the variation in the emission intensity of the phosphor sample with temperature during the heating and cooling processes, and the results are exemplified in Figure 8 [11,14,89]. It can be seen in Figure 8 that the emission intensity of a given phosphor at room temperature increases considerably after the heating–cooling cycles compared to the initial value of the pristine sample. The enhancement in emission spectral intensity at room temperature after heating–cooling cycles, in my opinion, likely stems from the change in the surface state of the phosphor sample, along with a change in the geometrical configurations of the spectrofluorometer arising from the irreversible change in the sample volume during the heating and cooling cycles. Because these Eu^2+^-activated phosphors were prepared by solid-state reactions at temperatures above 1200 °C in a reducing atmosphere, the possibility that the IQE of these phosphors is improved after undergoing heating and cooling cycles in ambient atmosphere in the spectral measurements is extremely low. This also suggests that the intensity enhancement in these phosphors at elevated temperatures likely arises from the same reasons, that is, NTQ; an increase in emission spectral intensity at high temperatures could originate from the change in the surface state of the phosphor sample or/and the change in geometrical configurations of the spectrofluorometer, not necessarily from an enhanced conversion efficiency of the phosphor. Therefore, temperature-dependent emission spectral intensity could not be a reliable measure for evaluating fluorescence TQ.

IQE that presents a direct measure for the efficiency of the conversion of absorbed photons into emitted photons is one of the key spectroscopic parameters of phosphors. The IQE of a phosphor is customarily measured by an integrating sphere setup by either a “two measurement” or a “three measurement” approach [90,91]. In the two-measurement approach, both an empty sphere measurement and a measurement with the sample in the sphere with the excitation beam incident on the sample are performed. In the three-measurement approach, an additional measurement with the sample in the sphere, but out of the excitation beam, is performed [91]. Unlike a spectrofluorometer, which only detects a certain fraction of the emitted light, an integrating sphere setup captures the total amount of light emitted from the excited sample or scattered by the sample and, hence, allows for the absolute measurement of the number of emitted photons and the number of absorbed photons. The number of absorbed photons follows from the decrease in the incident excitation light intensity (measured with a blank at the sample position) caused by the absorbing sample in the integrating sphere setup. Since the radiant power from the phosphor is geometrically distributed in a uniform manner due to multiple reflections, the integrating sphere setup is insensitive to the alignment of a beam and optical components, and, hence, the measurement results are highly repeatable.

*IQE* can also be determined by measuring the lifetime (*τ*), being the reciprocal of the sum of the rates of radiative decay (*Γ_R_*) and nonradiative decay (*Γ_NR_*) of the excited center if the lifetime without nonradiative decay (*τ*_0_) is known. Based on the definition of lifetime, Equation (4) could be modified as:(6)$$\;IQE=\frac{\tau }{\tau_{0}}$$

For fully allowed electric dipole transitions, like 4f^n−1^5d^1^ → 4f^n^5d^0^ transition of Eu^2+^ and Ce^3+^ ions, the radiative decay time is constant. In the case that nonradiative decay contributes to the transition rate from the excited state, luminescence lifetime decreases proportionally [52,63]. Hence, IQE could be easily obtained using Equation (6). However, the identification of the purely radiative decay rate must be considered. It is worth mentioning that for parity and spin-forbidden transitions, like ^2^*E*_g_ → ^4^*A*_2g_ transition of Mn^4+^, the radiative decay time in the order of milliseconds is not a constant. The spin–orbit coupling, which produces a non-zero contribution of the spin-doublet states in the spin-quartet states, allows the spin-forbidden transition to occur. The radiative lifetime of Mn^4+^-doped phosphors may change with the degree of spin–orbit coupling. I discussed this issue in a previous paper [73]. When investigating the IQE of Mn^4+^-doped phosphors using Equation (6), the change in radiative lifetime due to the variation in the spin–orbit coupling should also be taken into consideration.

The IQE of the phosphor could also be determined using a relative method, i.e., comparison of integral emission spectra of the sample and the standard under identical measurement conditions for samples of known absorption factors at the excitation wavelength. This requires not only a standard with known IQE and optical properties closely matching those of the investigated sample properties but also the measurement of the sample and standard under identical conditions [87]. Once the measuring conditions have changed, the validity and accuracy of the relative IQE could not be ensured [92,93,94].

Spectrofluorimetry is an important tool for phosphor research. Emission spectra can provide important information about the nature and energy of the emitting excited state. The shape of the spectra (narrow line or broad band) indicates the emission originating from either an allowed or a forbidden transition, while the centroid wavelength of the emission represents the energy difference between the ground and excited levels of the emitting centers. The full width at half maximum (FWHM) of the emission carries information regarding the uniformity of the surrounding environment of the emitting centers, and so on. However, emission spectral intensity gives little information about the conversion efficiency of the phosphor. If one hopes to study the conversion efficiency of a phosphor by comparing the relative emission spectral intensity of the sample with that of the reference standard, not only the measurement of the sample and standard should be performed under identical conditions but also the optical properties of the standard should closely match those of the investigated sample.

Luminescence TQ refers to the drop in conversion efficiency of the given phosphor due to the onset and enhancement of nonradiative decay at elevated temperatures. Investigating the TQ property of phosphors by measuring the temperature-dependent emission spectral intensity is based on an assumption that the variation in integral emission spectra with temperature could represent the variation in IQE with the temperature of the phosphor investigated. Unfortunately, the variation in the surface state and/or geometrical configurations associated with the volume change in the sample at high temperatures invalidates this assumption.

Quantitative luminescence measurements at varying temperatures are often not as easy as they seem at first glance, since the electric signal produced by a spectrofluorometer is related to the total luminescence intensity through a number of instrumental factors (intensity of the exciting source, instrumental optics, signal amplification) and can often hide artifacts. Many method-inherent problems are often neglected, resulting in measurements that are unreliable and of poor quality.

It is worth noting that the validity of experimental results derived from the temperature-dependent emission spectra depends on the information of interest. If the information of interest is the variation with temperature of the energy position of the emission, FWHM, or relative intensities of multiple peaks in a spectrum, uncertainty caused by changes in the surface state of the sample or geometrical configurations of the spectrofluorometer could be neglected once the spectrofluorometer has been calibrated with the standard light sources. These spectral characteristics could be derived from a power distribution profile of the emission on either an absolute scale or normalized scale. However, if the information of interest is the variation in conversion efficiency with temperature, the consistency of the measurement conditions matters. This requires a comparison of the absolute emitted intensity (total energy) of the sample measured in identical conditions. A quantitative comparison of the integral emission spectral intensity of a given phosphor measured at different conditions is of little scientific significance.

For a better assessment of the thermal stability of phosphors, it is advisable to measure the QE of the phosphor in question at different temperatures. If the temperature-dependent emission spectral intensity is employed as a measure for evaluating TQ performance of the phosphor, temperature-dependent excitation (absorption) spectra and decay time of the given phosphor should also be measured to check whether the associated data are self-consistent. From an application viewpoint, if a specific phosphor possesses NTQ, it should also be verified in the prototype WLED device under practical operating temperature (drive current). The temperature dependence of the emission spectral intensity alone is not a sound metric justifying the NTQ behavior of phosphors.

## 4. Summary

In recent years, many literature reports on the NTQ of various phosphors have been conducted. Due to space limitations, this article only discusses a portion of them. However, issues with measurement errors originating from a change in the surface state of the sample and geometrical configurations of the spectrofluorometer due to the change in sample volume are common. It should be emphasized that these NTQ conclusions are primarily based on the temperature-dependent emission spectral intensity of the phosphor under investigation, without paying attention to the contribution of extrinsic factors to the emission spectral intensity. This mindset of studying the TQ property of phosphors by measuring emission spectral intensity at different temperatures, perhaps, has a historical background. The researchers, therefore, seem to be unaware or unmindful of the fact that if the surface state of the sample and/or geometrical configurations of the spectrofluorometer change with temperature, the comparison of the integral spectral intensity at different temperatures could not reflect the variations with temperature in the conversion efficiency of the phosphor.

If the enhancement in emission spectral intensity originates from improved luminescence efficiency of the given phosphors at high temperatures, the research and development of these phosphors and unraveling the mechanism underlying are undoubtedly of great significance. However, this needs substantiation by either rigorous physical principles or practical applications. If the enhancement in the emission spectral intensity with temperature is due to measurement errors caused by changes in spectral measuring conditions, the experimental results are not credible, and the associated explanations for NTQ, such as defect-related energy transfer, increased phonon numbers at high temperatures, etc., are just beautiful legends.

This paper discusses the typical NTQ results reported in the literature from a macroscopic viewpoint based on common sense. This is hoped to arouse awareness in the phosphor community regarding issues with the papers reporting NTQ of the specific phosphors based on emission spectral intensity. It is specifically emphasized that when emission spectral intensity is employed to evaluate the conversion efficiency of phosphors, maintaining the same set of measurement conditions for the sample and reference standard is of vital importance. The measurement of emission spectra at varying temperatures requires a careful and detailed consideration of a broad range of physical concepts and practical instrumentation to produce an accurate, reproducible, and internationally acceptable result.

## Figures and Tables

**Figure 1 materials-17-00586-f001:**
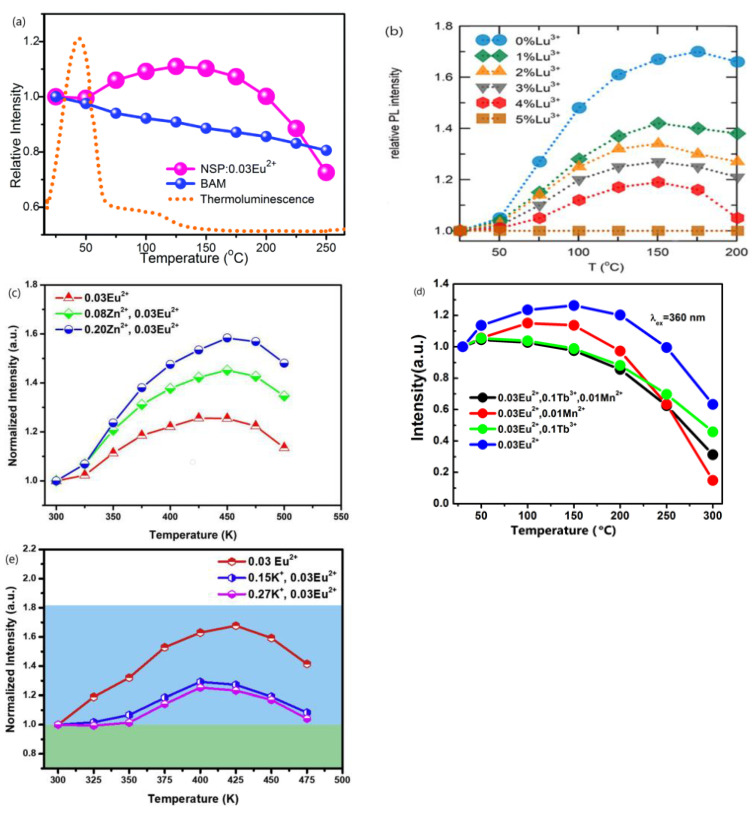
(**a**) Temperature dependence of the emission intensity of NSP:0.03Eu^2+^ compared with that of the commercial phosphor BAM. The dotted line represents the thermoluminescence curve of NSP:0.03Eu^2+^ [5]. (**b**) Temperature-dependent PL variation in the 450 nm peak of the NSPO:0.03Eu^2+^, yLu^3+^ phosphor as a function of Lu^3+^ concentration [1]. (**c**) Emission intensity of Na_3_Sc_2_(PO_4_)_3_:0.03Eu^2+^, Na_3_Sc_1.92_Zn_0.08_(PO_4_)_3_:0.03Eu^2+^, and Na_3_Sc_1.80_Zn_0.20_(PO_4_)_3_:0.03Eu^2+^ from 300 to 500 K; the initial emission intensities at room temperature are normalized to 1 [8]. (**d**) Temperature dependence integrated luminescence intensity of NSPO:0.03Eu^2+^ (blue line), NSPO:0.03Eu^2+^, 0.1Tb^3+^(green line), NSPO:0.03Eu^2+^, 0.01Mn^2+^ (red line) and NSPO:0.03Eu^2+^, 0.1Tb^3+^, 0.01Mn^2+^ (black line) [42]. (**e**) Temperature-dependent integrated intensity of NSP:0.03Eu^2+^, K_0.15_NSP:0.03Eu^2+^ and K_0.27_NSP:0.03Eu^2+^ [7]. Reproduced with permission from the respective references. Copyright 2016 The Royal Society of Chemistry, 2021 American Chemical Society, 2018 The American Ceramic Society and 2019, 2020 Elsevier B.V., respectively.

**Figure 2 materials-17-00586-f002:**
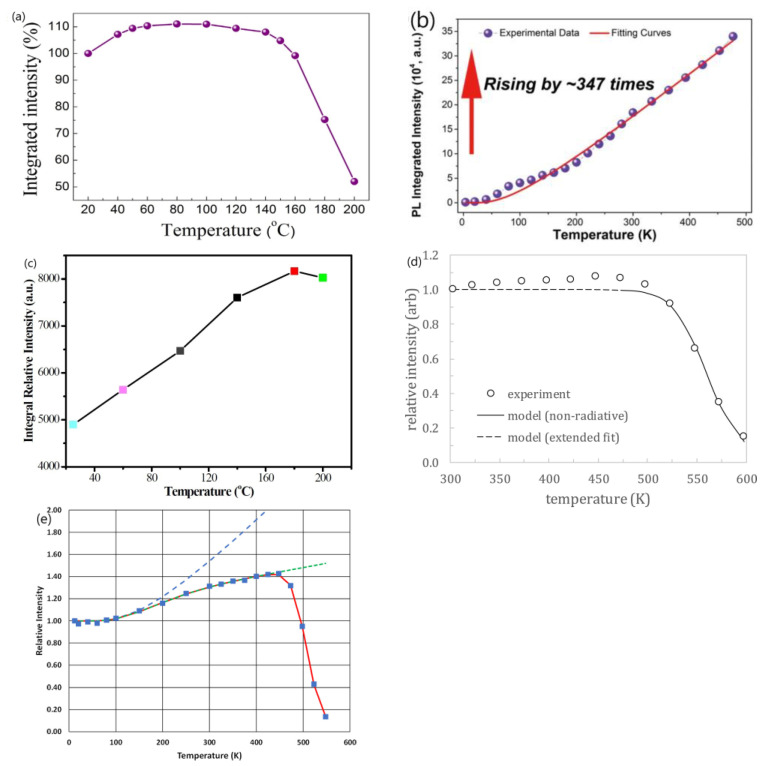
Temperature dependence of emission intensity of K_2_SiF_6_:Mn^4+^ phosphor reported by different researchers. (**a**) Ref. [27]. (**b**) Ref. [28]. (**c**) Ref. [30], and of TriGain^®^ K_2_SiF_6_:Mn^4+^ commercial phosphor (**d**) Ref. [59], (**e**) Ref. [25] (blue squares are experimental intensity). Reproduced with permission from the respective references. Copyright 2016 Elsevier B.V, 2018 American Chemical Society, 2015 The Royal Society of Chemistry, 2018 Elsevier, 2023, The Electrochemical Society, respectively.

**Figure 3 materials-17-00586-f003:**
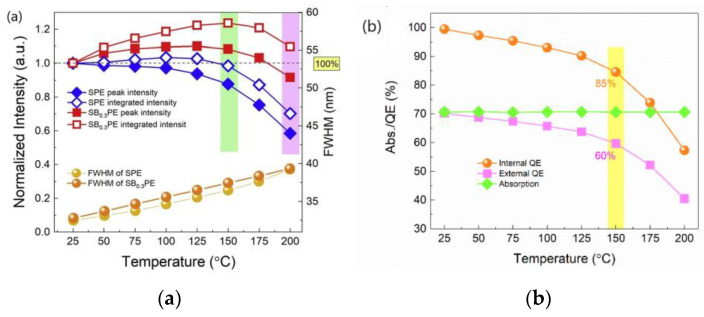
(**a**) The relationship of peak and integrated intensity, FWHM of SPE, and SB_0.3_PE versus different temperatures. (**b**) The variations in IQE, EQE, and absorption efficiency versus temperature from 25 °C to 200 °C of SB_0.3_PE phosphor [11]. Reproduced from Ref. [11]. Copyright 2020 The Author(s).

**Figure 4 materials-17-00586-f004:**
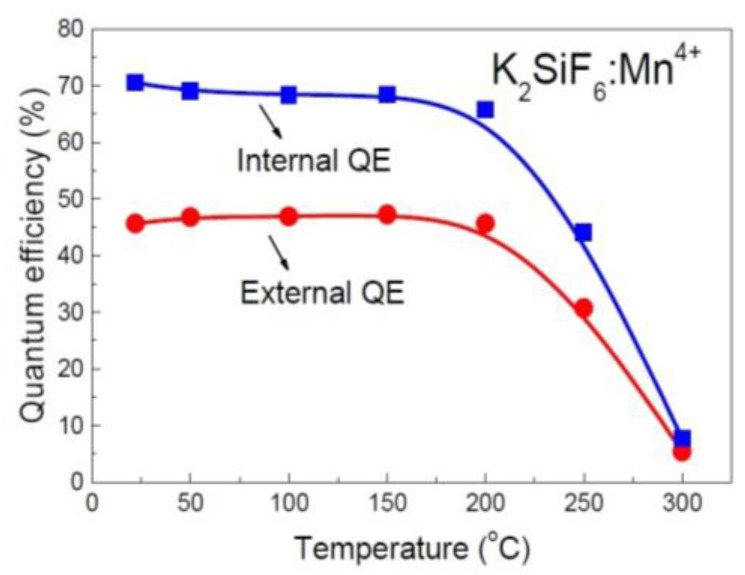
Temperature-dependent IQE and EQE of K_2_SiF_6_:Mn^4+^ when excited at 450 nm [53]. Reproduced from Ref. [53] with permission, Copyright 2015 Optical Society of America.

**Figure 5 materials-17-00586-f005:**
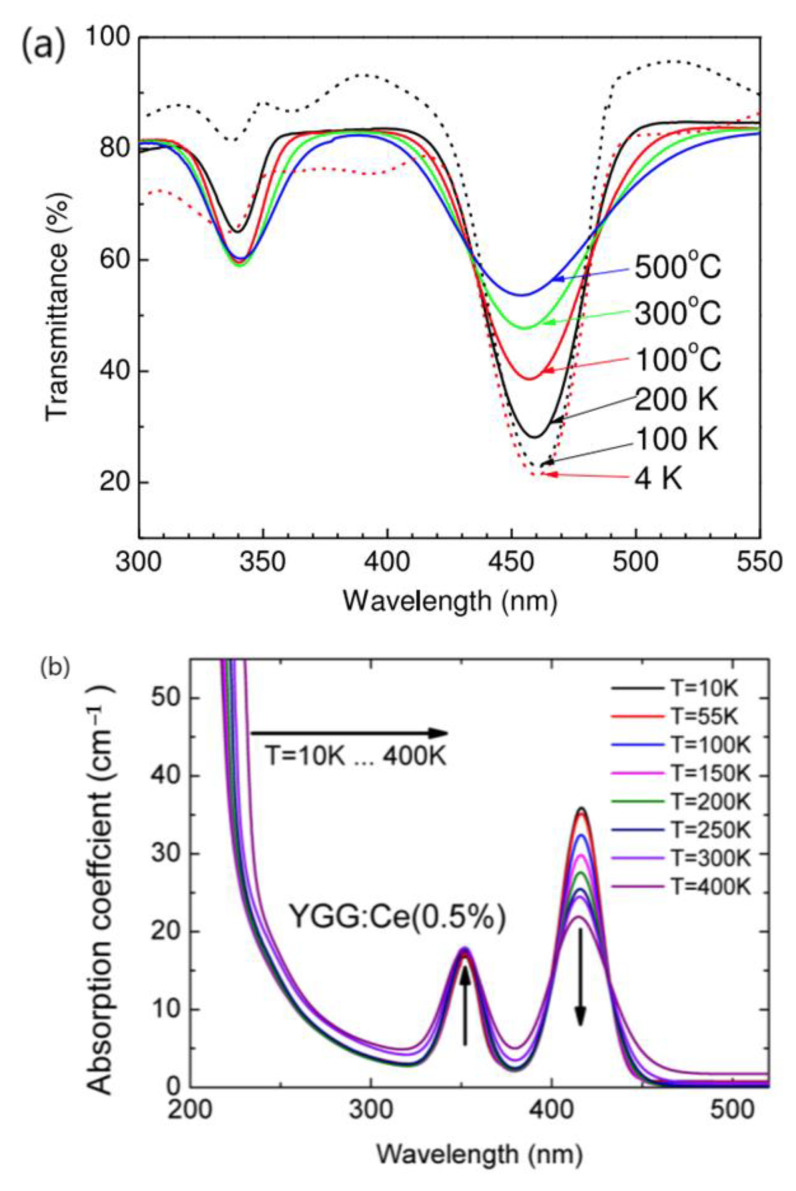
(**a**) Transmittance spectra of Ce:YAG from low to high temperature [80]. (**b**) Temperature dependence of the UV and VIS absorption spectra of the YGG:Ce (0.5%) crystal [83]. (**c**) Excitation and emission spectra of La_2.94_Ce_0.06_Si_6_N_11_ synthesized at 1950 °C for 2 h under nitrogen atmosphere of 0.92 MPa [84]. Reproduced with permission from respective references, Copyright 2015 IOP Publishing Ltd., 2015 Optical Society of America, 2009 ECS–The Electrochemical Society, respectively.

**Figure 6 materials-17-00586-f006:**
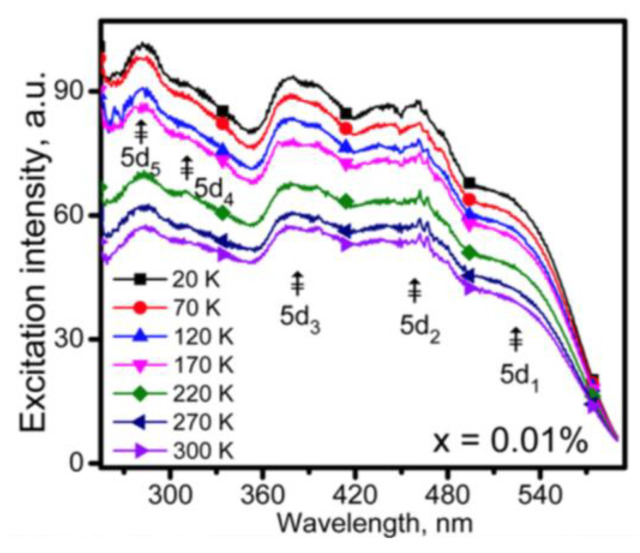
Excitation spectra of Ca_0.99_Eu_0.01_AlSiN_3_ measured at various low temperatures [85]. Reproduced with permission from Ref. [85], Copyright 2016 American Chemical Society.

**Figure 7 materials-17-00586-f007:**
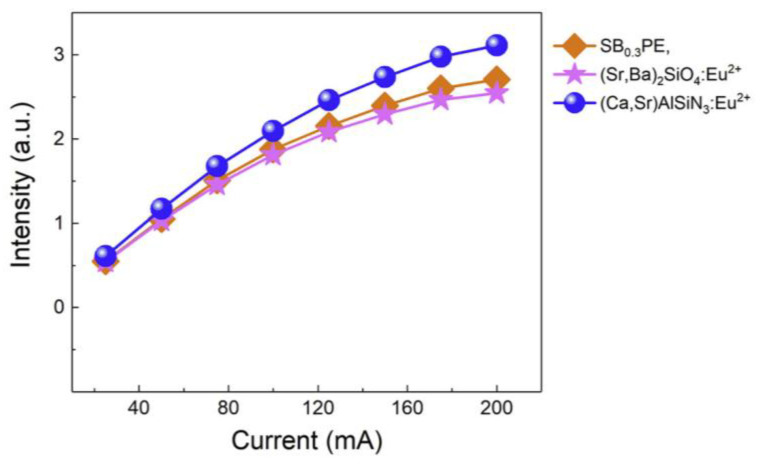
The emission intensities of SB_0.3_PE blue, (SrBa)_2_SiO_4_:Eu^2+^, and (SrCa) AlSiN_3_:Eu^2+^ phosphors as a function of driving current [11]. Reproduced from Ref. [11]. Copyright 2020 The Author(s).

**Figure 8 materials-17-00586-f008:**
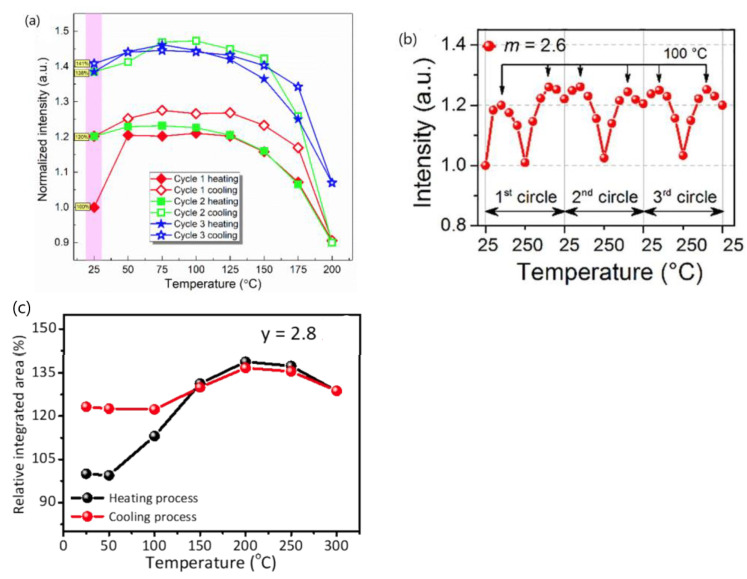
(**a**) Thermal degradation tests of SB_0.3_PE ((Sr_0.69_Ba_0.3_)_2_P_2_O_7_:0.02Eu^2+^) phosphor by heating and cooling phosphor for 3 times [11]. (**b**) Normalized integrated PL intensity versus temperature with 3 heating–cooling circles of K*_m_*_−0.4_Al_11_O_17+δ_:0.2Eu^2+^ (*m* = 2.6) [14]. (**c**) The correlation between the heating and cooling process of the Na_2.87_Sc_2_(PO_4_)_2.8_:0.13Eu^2+^ phosphor by the relative-integrated area series [89]. Reproduced with permission from respective references. Copyright 2020 The Author(s), 2021 Elsevier B.V., and 2022 The Royal Society of Chemistry, respectively.

**Table 1 materials-17-00586-t001:** Chromaticity coordinate and CCT of the WLED devices fabricated using specific phosphors exhibiting NTQ in spectral measurement under different drive currents ^a^.

No	Composition of WLED	Drive Current/mA	Chromaticity Coordinate		CCT/K	Ref.
			x	y		
**1**	Blue chip (λ = 455 nm) + YAG 04 + **K_2_TiF_6_:Mn^4+^**	20	0.4575	0.4124	2748	[29]
		40	0.4588	0.4160	2757	
		60	0.4569	0.4158	2783	
		80	0.4539	0.4152	2821	
		100	0.4550	0.4172	2820	
		120	0.4534	0.4158	2833	
**2**	Blue chip + YAG:Ce^3+^ (0.1 g) + **K_3_AlF_6_:Mn^4+^** (0.4 g)	40	0.4208	0.4014	3270	[68]
		120	0.4178	0.3951	3275	
		240	0.4142	0.3874	3284	
		300	0.4127	0.3832	3278	
**3**	UV chip (λ = 365 nm) + **Na_3_Sc_2_(PO_4_)_3_:0.03Eu^2+^** + La_3_Si_6_N_11_:Ce^3+^ + (Sr,Ca)AlSiN_3_:Eu^2+^	100	0.2989	0.3502	7041 ^b^	[2]
		200	0.2968	0.3531	7116	
		300	0.2958	0.3574	7121	
		400	0.2952	0.3602	7121	
		500	0.2953	0.3617	7101	
		600	0.2957	0.3625	7075	
		700	0.2963	0.3632	7040	
		800	0.2970	0.3633	7006	
		900	0.2975	0.3639	6978	
		1000	0.2982	0.3639	6945	
**4**	UV chip (λ = 365 nm) + **Sr_1.38_Ba_0.6_P_2_O_7_:0.02Eu^2+^** + (SrBa)_2_SiO_4_:Eu^2+^ + (Sr,Ca)AlSiN_3_:Eu^2+^	25	0.3958	0.4065	3831	[11]
		50	0.3957	0.4050	3823	
		75	0.3957	0.4036	3814	
		100	0.3956	0.4028	3811	
		125	0.3955	0.4019	3807	
		150	0.3955	0.4008	3799	
		175	0.3956	0.3997	3789	
		200	0.3958	0.3984	3775	

^a^: The formula of the phosphor which showed NTQ in spectral measurement is highlighted in bold. ^b^: CCT values are calculated from the chromaticity coordinates x and y given in Supplementary Table S5 of Ref. [2] according to McCamy’s formula [70]: *T* (K) = −449*n*^3^ + 3525*n*^2^ − 6823.3*n* + 5520.33 where *n* = (x − 0.3320)/(y − 0.1858).

## Data Availability

No new data were created.

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
