# Peer review of "Negative Thermal Quenching of Photoluminescence: An Evaluation from the Macroscopic Viewpoint"

_materials, 2024, doi:10.3390/ma17030586_

Round 1
Reviewer 1 Report
Comments and Suggestions for Authors The present paper studies in detail the quenching processes of phosphorluminescence through an exhaustive comparison with the literature. Even if the manuscript does not report original results,
the content is certainly interesting for a wide audience that uses luminescence
variation as a crucial property for many applications, first of all sensing.
Overall, the work deserves to be accepted,
although the author could address some points to improve
the clarity of the manuscript.
- the results reported in Table 1 should contain uncertainty,
both experimental and analytical. This is crucial for the reproducibility
of the proposed method.
- as specified in Eqs. 4, 5 and 6, the quantum efficiency can be measured by the
lifetime variation, as long as the purely radiative decay is identified.
The author could discuss this point better and make a comparison with some
literature data to interpret the results reported in this work.
Author Response
Please see the attached word file for the detailed reply to the review report.

Reviewer 2 Report
Comments and Suggestions for Authors
The manuscript “Negative Thermal Quenching of Photoluminescence- An Evaluation from Macroscopic Viewpoint” authored by Shirun Yan, is a nice piece of paper. The thoughtful reflection presented in section 3 is very important and consistent for the luminescence community of solid materials working in all areas of research, not only NTQ, and in my opinion, it deserved a broader title. The section 2, however, seems disconnected sometimes, and the same care in selecting examples was not taken. I have raised a few points for consideration, but I consider this paper to be fine for publication at Materials.
Line 105-110: The author points out that the Tmax, regardless of the scale is in Celsius or Kelvin, is different in the different reports, however, the temperature is about the same in all shown papers, within a range of 30 degrees (150 C-177C), although the intensity varied quite a lot. What author considers a significant temperature change?
Line 239-245, and Eq (1): Is not the Ess energy, accounted as non-radiative energy, already counted in the second term on the right-hand side of Eq(1)?
The question posed by authors in lines 170-171 is not answered in item 2.2. The conclusion drawn in lines 303-305 does not allow the readers to conclude anything on energy conservation. It is not clear what is the aim of this question.
Regarding item 2.3: Did the authors consider the effect of photostability of the different phosphors when discussing this topic? It is known that different phosphors have different temporal stability. For the four examples cited, were conducted any photostability measurements, or did the authors in the references check the reversibility of the process? Is the change in CCT permanent or reversible? I am afraid there are more things behind the results reported.
In pg 17 do the author mean IQE or EQE? The definition presented is more consistent with EQE rather than IQE.
Comments on the Quality of English LanguageA few mistyping were observed. The quality is very good.
Author Response
Please see the attached word file for the detailed reply to the review report .

Reviewer 3 Report
Comments and Suggestions for Authors
Dear Aurhors, you have done a great job.
Comments on the present Review
The present paper contains a review on investigations, containing experimental results and their evaluation on an actual phenomenon – the so called negative thermal quenching (NTQ) of photoluminescence (PL), characteristic of several essential prospective complex phosphors doped with rare earth ions (Eu2+) or transition metal ions (Mn4+). The main of them are NSP:Eu2+, KSF:Mn4+, BAM and others.
A Review is focussed on gathering and evaluation of information of actual problems, essential for application of appropriate materials, based on assessment of the relevant literature. (The list of References contain 122 items). The main questions, which are analyzed in this Review, are the following.
1.Are the NTQ measurements in the given materials reproducible? The NTQ properties of Na3Sc2(PO4)3:Eu2+ and Mn4+- doped fluorides are discussed, which are based on the results of PL dependence on temperature, shown in Figs. 1 and 2, correspondingly. In my opinion, there is also necessary to present the PL and PLE (excitation) spectra of these materials, which are important for better undertanding of spectral conditions. They are also necessary for the discussion of the next question, including the luminescence mechanisms. Besides, important is the thermoluminescence (TL) grafs, which is is only given for NSP:Eu material (Fig. 1 a). Information on the bandgap energy of materials would also be desirable.
As a result of the analysis, the authors come to the acceptable conclusion, that NTQ phenomenon of above mentioned materials is reproducible within a resonable range of measurement errors.
2. How the law of conservation of energy is fulfilled in NTQ materials? This question involves the mechanisms of the NTQ phenomenon.
I have several important objections to the presentation of the issue.
As is known, luminescence is basically characterized by intra-center and recombination mechanisms. In the case of intra-center luminescence, all optical processes related to absorption and emission, occur in the same luminescent defect (dopant ion). In the case of recombination mechanism, besides the luminescent ion, several other defects are also involved, including the trapping centers for electrons and holes.
To describe the luminescence processes in the mentioned materials, the authors use the equation (1) (186), which directly describes the intra-center luminescence. All separate parts of the equation are analyzed correctly, but I cannot accept the explanation of the NTQ phenomenon only by intra-center processes without including participation of other centers in the luminescence process.
At (177) the Authors maintain: “ Considering that no afterglow luminescence was observed at any temperature in the phosphors reported with NTQ, it means that the light absorption and emission processes do not involve energy storage and a delayed release for these phosphors” .
I disagree with that statement. There are several experiments, which demonstrate a presence of energy storage of the material, caused by light irradiation. Beside the observation of luminescence afterglow or appearance of the so-called persistent luminescence (PersL), an energy storage is also detected through presence of thermoluminescence (TL) and measurements of electron paramagnetic resonance (EPR). For NSP:Eu material the TL curve is shown in Fig 1, a (132). From Ref. [5] it is known, that TL and NTQ measurements are realized at the same wavelength of exciting light at 365 nm. This fact proves, that irradiation of NSP:Eu material with 365 nm results in both - the Eu direct intra-center excitation together with the energy storage caused by charging or recharging of trap centers with electrons or holes and others. As known, intensity of PersL is dependent on temperature and its absence in above material, mentioned by the Authors, can be related to its low intensity at room temperature (RT). From Fig. 1, a it is seen, that the main maximum of TL peak is at ≈50 0C, where the release of charged particles from the trap centers is with the most efficiency, thus possible increasing the intensity of the PersL.
Besides, irradiation of NSP:Eu with 365 nm light does not result only in excitation of Eu ions, because the PLE band is wide and obviously complex (Fig. 4 from [5]). Therefore, excitation of unknown defects with their ionization is also possible.
In my opinion, the authors should rework-transform this section of the Review, including also recombination luminescence processes.
3. The third part - could NTQ of a given phosphor be demonstrated in prototype WLED device?
This part of the Review is well written and contains a valid information.
In summary. In my opinion, it is a valuable overview of series of studies of NTQ phenomenon in solid state luminescent materials, and it can be published after some substantional improvements-additions, discussed above.
Best regards,
Reviewer.
